# An Investigation on Microstructure, Texture and Mechanical Properties of AZ80 Mg Alloy Processed by Annular Channel Angular Extrusion

**DOI:** 10.3390/ma12061001

**Published:** 2019-03-26

**Authors:** Xi Zhao, Shuchang Li, Yong Xue, Zhimin Zhang

**Affiliations:** 1School of Mechanical and Electrical Engineering, North University of China, Taiyuan 030051, China; s1701079@st.nuc.edu.cn; 2School of Materials Science and Engineering, North University of China, Taiyuan 030051, China; yongxue395@163.com (Y.X.); zhangzhimin@nuc.edu.cn (Z.Z.)

**Keywords:** AZ80 Mg alloy, annular channel angular extrusion, microstructure, texture, mechanical properties

## Abstract

Annular channel angular extrusion has been recently developed as a new single-pass severe plastic deformation method suitable for producing large size cup-shaped parts from cylindrical billets. In this study, the novel technology was successfully applied to commercial AZ80 Mg alloy at 300 °C, and microstructure, texture evolution, and mechanical properties were investigated. Due to severe shear deformation, the initial microstructure, including the coarse grains and large eutectic β-phases, was greatly refined. The strong basal texture formed during the initial deformation stage was modified into a weak tilted dynamic texture. During the deformation process, fine β-particles separated from eutectic phases effectively hindered the grain boundary migration and rotation, enhancing the grain refinement and texture weakening. More than 63% of the microhardness increase was achieved in this extruded part. Also, tensile tests showed the yield strength and elongation in both directions (transverse and longitudinal) of extruded part were improved more than 2.5 times, and the ultimate tensile strength was improved more than 2 times, compared to the initial material state. The improved material properties were mainly attributed to microstructure (grain and phase) refinement and texture weakening. It was demonstrated that the annular channel angular extrusion process can be considered as a novel and effective single-pass severe plastic deformation method.

## 1. Introduction

Magnesium (Mg) alloys, being among the lightest structural materials, have become a focus in the world due to their advantages [1,2], especially the high specific strength compared to other alloys [3,4]. In the past twenty years, Mg alloy structural components have been widely applied in automotive, aerospace, defense, and electronic industries, while more than 90% of them are in the as-cast state [5]. With the higher requirement of material strength, casting Mg alloy products have gradually failed to meet practical application needs. Therefore, large-scale development of forged Mg alloys will be an important avenue for the future development of Mg alloy materials.

In general, the mechanical properties of most metals can be effectively improved by modification of their microstructure via alloying, processing, heat treatment, and so on. Therefore, typically forming processes such as rolling, forging, and extrusion have been extensively employed to process Mg alloy materials [5,6]. However, conventional forming methods often fail to achieve significant refinement of microstructure and improvement of mechanical properties due to lower strain levels. Besides, due to the hexagonal close-packed (HCP) crystal structure of Mg, conventional deformation methods usually also induce a pronounced crystal texture (basal texture), which dramatically reduces the ability of basal slip during subsequent deformation [7,8]. This would adversely affect the application of final products. In this respect, the most severe plastic deformation processes (SPD) show unparalleled advantages in improving the mechanical properties of Mg alloys due to their excellent microstructure refining ability and texture modification ability. A fine grain structure, accompanied with tilted shear texture, has already been widely reported in Mg alloys deformed through equal channel angular process (ECAP) [9,10]. Wang et al. [11] systematically investigated the capacity of cyclic extrusion and compression (CEC) in improving the microstructure, mechanical properties, and developing nonbasal texture, which included AZ31, AZ61, and AZ91 Mg alloys. Similarly, Maghsoudi et al. [12] also investigated the effect of accumulative back extrusion (ABE) process on the AZ81 Mg alloy, and revealed that applying the ABE process up to five passes led to simultaneous modification of the phase morphology, grain structure, as well as achieved texture weakening of the material. However, the current SPD methods are only suitable for bulk material preparation and fail to provide industrial-scale samples. In addition, due to the multipass forming process, the production efficiency is also greatly limited. These factors seriously hinder their further application, and thus further study is still needed before their industrialization.

In recent years, annular channel angular extrusion—a new type of single-pass SPD technology that supplements ECAP into traditional backward extrusion (BE) deformation [13]—is attracting more and more attention. Similar to the BE method, the new method can be used to produce close-ended products in one pass and does not require subsequent operation. Compared to the multipass forming method, it seems to be a very promising SPD process that can be applied to industrial production. Shatermashhadi et al. [13] reported the new method through finite element (FE) simulation, and revealed that the novel process could obtain more than twice the effective strain than the traditional BE method. To obtain the minimum pressing force during deformation, Hosseini et al. [14], through use of the upper bound method, calculated the optimum ratio of height to diameter of billet. Also, the feasibility of the process has been verified by the production of small-sized pure aluminum (Al) test part, which revealed significant grain refinement and performance improvement [15]. However, further application of the process to achieve plastic forming of alloy materials, particularly Mg-based alloys, has not been reported. Undoubtedly, microstructural refinement and deformation texture weakening by SPD deformation have an important influence on improving the mechanical properties of Mg alloy materials.

In this study, the commercial AZ80 Mg alloy was successfully applied to the annular channel angular extrusion process. The aim of this work is to investigate the effect of the novel BE process on the microstructure, texture, and mechanical properties of the applied AZ80 Mg alloy, respectively.

## 2. Materials and Methods

A commercial AZ80 Mg alloy (Mg-8.0Al-0.5Zn-0.11Mn, wt.%) (1000 mm in length and 100 mm in diameter) was prepared by Wenxi YinGuang of Magnesium Industry (Group) Co., Ltd. (Wenxi, China). Then, a cylindrical billet 360 mm in height and 90 mm in diameter was cut from the middle of the as-cast rod. The annular channel angular extrusion process was conducted with a series of composite mold, the schematic diagram of the mold is shown in Figure 1. The detailed operating method has been reported by Shatermashhadi et al. [13]. Prior to extrusion, the billet was preheated at 300 °C for 3 h to enhance its deformability and reduce the risk of cracking. The extrusion was then conducted at 4-THP-630 hydraulic press with a punch speed of 1 mm/s (the MoS_2_ paste was used for lubrication). Finally, a good quality part, with an outer diameter of 200 mm, an inner diameter of 160 mm, and a height of ~160 mm, was produced.

The microstructure of the extruded part were observed using optical microscopy (OM, A1m, Zeiss Inc., Oberkochen, Germany), scanning electron microscopy (SEM, SU5000, Hitachi Inc., Tokyo, Japan), and an electron backscattering diffraction apparatus (EBSD, EDAX Inc., Mahwah, NJ, USA) installed in the SEM. Specimens for OM and SEM analysis were prepared by mechanical polishing and etching using a solution of oxalic acid (0.35 g), glacial acetic acid (2 mL), and distilled water (58 mL). Grain size and phase fraction were measured using the software Image-Pro Plus 5.0 (Media Cybernetics Inc., Bethesda, MD, USA). For EBSD examinations, the specimens were ground and electropolished in the electrolyte (90% C_2_H_5_OH + 10% HClO_4_) at 15 V for ~30 s at −35 °C. EBSD was conducted with a scan step length of 0.5–0.6 μm. The EBSD data were analyzed by the Orientation Imaging Microscopy (OIM) Analysis-v7.3 software (EDAX Inc., Mahwah, NJ, USA). In OIM software, the misorientation angle quantification was used to identify low angle grain boundaries (LAGBs, 2°–15°) and high angle grain boundaries (HAGBs, 15°–180°), as indicated in the EBSD maps by white and black lines, respectively. Meanwhile, in order to analyze the recrystallization (DRX) behavior, the grain orientation spread (GOS) was applied to differentiate between deformed grain and DRXed grain [16]. In this work, the DRXed grain was identified by a GOS value smaller than 2°, while the deformed grain had a GOS value greater than 2°.

Tensile test was carried out using an Instron 5967 machine (Instron Inc., Canton, MA, USA) with an initial strain rate of 1 × 10^−3^ s^−1^ at room temperature. Tensile specimens, whose gauges were 15 mm in length, 4 mm in width, and 2 mm in thickness, were extracted from the wall of the extruded part, and three experiments were conducted for each condition. Microhardness along the deformation routine was performed on a Vickers indenter with a load of 200 g as well as a loading time of 15 s, and at least 10 different points in each area were measured.

The DEFORM 3D-v11 software (SFTC Inc., Columbus, OH, USA) was used for FE simulation. In this study, the employed material was the AZ80 Mg alloy. The material model was defined as elastic-plastic. The billet was established by simple geometry in Deform. All mold parts had been defined as rigid bodies, and were drawn by introducing simple 2D section rotation. Set the punch down speed to 1 mm/s. The friction between the workpiece and the matrix was shear friction, and the friction coefficient was 0.3. In order to ensure the relative accuracy of the simulation, many simulations were done with 20,000 to 40,000 elements (2000 elements per group). The average effective strain distribution along a certain path in wall zone was then analyzed for each group. When the number of elements increased to 32,000, the strain distribution converged to a relatively stable range. Therefore, the billet was meshed using 32,000 elements and considered as an available model. Table 1 shows the parameters of the simulation.

## 3. Results and Discussion

### 3.1. Initial Material State

As shown in Figure 2a, the microstructure of the as-cast AZ80 Mg alloy exhibits a typical dendritic structure that is composed of an Mg matrix, coarse reticular eutectic β (Mg_17_Al_12_) phases, lamellar β-precipitates, and a small amount of Al_8_Mn_5_ phases [17]. Among them, the lamellar β-precipitates are mainly distributed around the eutectic phases, which are of typical morphology of discontinuous precipitate [17]. After the billet was reheated, as is seen in Figure 2b, the discontinuous precipitates are redissolved into the Mg matrix while a large number of rod-like β-precipitates precipitated around the eutectic phase boundaries. These precipitates are generally considered to be typical continuous precipitates in the Mg–Al alloy [18]. Unlike the sparse lamellar β-precipitates, it can be seen that these closely packed rod-like β-precipitates tend to precipitate along a certain direction, which indicates a certain orientation relationship with Mg matrix. This certain orientation relationship has also been reported in depth by many studies [18,19,20]: (0001)_m_//(110)_p_ and [-12-10]_m_//[1-11]_p_. The appearance of the precipitated precipitates after preheating indicates that the material has exhibited a slight aging behavior. This is fully understandable because the relatively low holding temperature of 300 °C.

### 3.2. Finite Element (FE) Analysis

The predicted flow grid and effective strain distribution were implemented to reveal the deformation characters of the part during the novel process because they can provide accurately observe the deformation behavior of the part. It can be seen from the flow grid distribution in Figure 3a that the part can be roughly divided into the deformation zone A, the first shear zone (SZ1), the bottom deformation zone B, the second shear zone (SZ2), and the wall zone C. The deformation zone A is formed in the upsetting stage, subjected to smaller deformation variables, and the mesh is hardly deformation. The grid of the first corner zone is offset from the horizontal grid by an angle that is related to the corner of the mandrel. The grid shape changes from rectangle to parallelogram, which implies the appearance of shear deformation. The grid has not been significantly affected in zone B. After the billet enters the second shear zone, the mesh is progressively gathered by the shear deformation. The bending angle of each line is different, which leads to further deformation and elongation of the grid in zone C. Corresponding to the flow grid distribution, as is seen in Figure 3b, the effective strain increases by more than 2.0 when billet passes through the first shear zone. The strain has not been improved notably in the bottom of the channel. After the billet passes through the second shear zone, the effective strain is further increased and the average value is over 3.3 in the wall. Such high effective strain is rarely reported at the end of the single-pass SPD processes. Also, the obtained effective strain is more than twice that of conventional BE method [21]. Therefore, when the material passes through the second shear zone, the higher effective strain of the novel process may significantly refine the microstructure and improve the material properties.

### 3.3. Effect of Extrusion on Microstructure

To explore the effects of different stress states on material deformation and to further reveal the microstructure evolution of materials under this new SPD method, we selected three typical zones (zones A and C: TD (transverse direction)–ND (normal direction) plane; zone B: LD (longitudinal direction)–ND plane) from the extruded part and performed microstructure observations on them. The results are discussed separately as follows.

The typical microstructure of zone A from the part is presented in Figure 4. Obviously, the microstructure in this initial upsetting stage is not adequately influenced by the deformation. As shown in Figure 4a–c, the fine DRXed grains more frequently occur at the boundaries of coarse grain and eutectic β-phase, resulting in the formation of a typical necklace grain structure. The average DRXed grain size and DRX fraction were measured at ~6.1 μm and ~23%, respectively (Table 2). In general, the appearance of necklace DRX structure is considered to be a symbol of continuous dynamic recrystallization (CDRX). It is related to a recovery process with dislocation rearrangement to form subgrains and continuous absorption of dislocations into LAGBs, which eventually leads to the formation of HAGBs and new DRXed grains [22]. More evidence can be seen from Figure 5a,b (the high magnification of Area 1 in Figure 4c); there are lots of LAGBs within the coarse grains and the subgrains defined partly by LAGBs (pointed at by blue arrow) and partly by HAGBs (pointed at by red arrow), which is the representative feature of CDRX. However, the observed fine grains formation at the vicinity of eutectic β-phase reveals that the coarse eutectic phase also plays a non-negligible role in the DRX process. This shows that under deformation the strain incompatibility develops at the interface between the harder phase and matrix, and the high density of dislocations forming around the phase provides a strong driving force for grain nucleation. This promotes nucleation of new grains and accelerates the DRX behavior is obviously a typical particle stimulating nucleation (PSN) phenomenon. It has also been reported during hot deformation of many phase containing Mg alloys, such as AZ31 + 0.8%Sr, M1-1.6Sr (Mg-Mn-Sr), Mg-4.3Li-4.1Zn-1.4Y [23,24,25], and so on. Moreover, it is also worth noting that the boundaries of the eutectic β-phase have begun to become serrated and, also, some small β-particles are separating from above. In the meantime, with the appearance of fine grains, the continuous β-precipitates have also begun to be divided into small parts. Compared with the coarse eutectic β-phases, these generated fine β-particles may not act as a PSN effect because of small size. However, their presence at the grain boundaries may hinder the migration and coalescence of grains by pinning effect, and thus can greatly delay the growth of newly DRXed grains [26].

Besides, in addition to the appearance of the necklace DRX behavior from the coarse grain boundaries, Figure 4c also shows that slight twin dynamic recrystallization (TDRX) has also occurred within the deformed grains at this stage (inside the white dotted circle). The observed twins include the {10-11} compression twining, {10-12} tension twining, and {10-11}-{10-12} double twining, in which their basal planes are tilted ~38°, ~86°, and ~56° with respect to the deformed grain, respectively [27]. It is well established that as a common deformation behavior of Mg and its alloys, the twining holds a crucial role in performing the c-axis strain and hence coordinate plastic deformation. It can be inferred that, at the current stage, due to the presence of large deformed grains having nearly basal orientation, the basal and the prismatic slip system cannot be sufficiently activated. Additionally, considering the relatively low deformation temperature, the pyramidal <c+a> slip system may also not be fully utilized. Therefore, in order to further coordinate the intragranular plastic deformation, the twin nucleation may occur subsequently.

The novel BE process is associated with shear deformation in corner zones. As shown in Figure 6b, due to the first shear deformation at the bottom of the mold, the eutectic β-phases have been significant deformed and elongated, and obvious fractures are also observed. The elongation of coarse eutectic β-phases during the deformation process indicates that they have softened and become deformable at the relatively high deformation temperature of 300 °C. At higher magnification, as shown in Figure 6b, it can be seen that the elongated phase has become locally weakened and is being completely torn apart into smaller parts under shear stress. It is obvious that these fine β-particles distributed on the grain boundaries will greatly limit the growth of fine grains by pinning effect. On the other hand, it can be seen in Figure 6a,c that a more complex grain structure, consisting of numerous fine DRXed grains, elongated deformed grains, and small number of coarse DRXed grains, is also generated in this stage. Among them, most of deformed grains (marked by the crystal in Figure 6c) keep their basal planes nearly parallel to TD, which is not conducive to basal slip. Also, a large amount of fine grains surrounding the deformed grains form local fine grain bands, and some smaller fine grain bands also occur inside or traversing the deformed grains. It is clear that the present deformation character shows a local DRX behavior caused by the local plastic deformation. Fatemi-Varzaneh et al. [21,28] reported the appearance of local deformation phenomenon and revealed the local DRX behavior when exploring the influence of ABE process on the microstructure of AZ91 and AZ31 Mg alloys. Due to the shear deformation acts on the material, the material tends to be unable to withstand the applied shear strain and thus the continuous plastic flow cannot be established swiftly. After that, the deformation is easy to follow in narrow regions of inner localized plastic flow, which leads to the formation of local deformation band (shear band) [21]. When further strain is imposed, continuous dislocation accumulation within the shear band accelerates local DRX behavior, which leads to the formation of the local fine grain band [28]. The currently observed fine grain bands and the finer grain bands that are occurring inside the deformed grain obviously originate from the local plastic deformation. Such deformation character accelerates to the formation of local shear band from coarse grain boundaries or interior, which leads to significant grain refinement.

When shifting to zone C, as shown in Figure 6e, the elongated eutectic β-phases have been further refined and more β-particles are separated from them simultaneously. As expected, the grains in regions with densely β-particles distribution still maintain small size. Table 2 shows the relative fraction of eutectic β-phase and fine β-particle at different stages. It can be seen that the relative fraction of the fine β-particle increases from 11% of the initial stage to 61% of the zone C. It is shown that the refinement of the eutectic phase is promoted remarkably by the introduction of strong shear deformation. Furthermore, unlike the previous deformation stage, it can be seen that most of the elongated deformed grains are also obviously refined into coarse DRXed grains, and form coarse grain bands, which are alternately distributed with fine grain bands. Therefore, a mixed grain structure, consisting of numerous fine DRXed grains, coarse DRXed grains, and a small amount of residual deformed grains, appears in the formed zone. Due to the significant grain refinement and the appearance of coarse DRXed grains, the average grain size of the extruded part is eventually refined to ~9.6 μm, while the average DRXed grain size increases to ~8.3 μm (Table 2). The observed formation of coarse DRXed grains within deformed grains can be seen in Figure 7 (the higher magnification of area 1 in Figure 6f). Clearly, the significant CDRX (pointed at by arrow) is occurring inside the deformed grain at this zone. Further, compared with most of the near basal orientation’s deformed grains in the bottom channel zone B (Figure 6c), new grains being formed (adjacent grains are divided by LAGBs), or those that have generated within the deformed grain show that their basal planes have been rotated away from the basal orientation. This shows that the basal slip and possible pyramidal <c + a> slip have been greatly activated inside the deformed grains compared to the bottom zone B (the orientation between them also changes by the rotate of prismatic due to the prismatic slip, but to a very low degree) [16,29]. In fact, when deformation enters the second corner zone, the principal stress axis will be changed during deformation due to the stress state changes from extrusion stress in bottom zone to shear stress. Then the change of principal stress axis will effectively alter the basal Schmid factor (SF_basal_) distribution of grains and thus lead to reactivation of the basal slip system [30]. On the other hand, due to the high shear stress in the corner zone, the pyramidal <c + a> slip system may also be activated effectively. Therefore, in the case where the basal and <c + a> slip are activated, dislocation energy storage within the deformed grains will accumulate rapidly with subsequent shear deformation. When these grains become thin enough and accumulate enough dislocations, it can be inferred that they will not only be consumed into fine DRXed grains from the boundary zones, but also numerous basal dislocations joining other possible dislocations (nonbasal dislocations) will promote the formation of intragranular LAGBs by the dynamic recovery (DRV) process. After that, DRXed grains are formed by the continuous evolution of the subgrains with the growth of LAGBs to HAGBs. Undoubtedly, the subsequent appearance of coarse DRXed grain bands leads to the grain refinement in the final stage and the formation of a more complete DRX structure.

### 3.4. Texture Evolution

The texture features from the three typical areas of the part were selected to reveal the texture evolution of the deformed part. In order to further clarify the relationship between texture and grain structure change, we distinguished the DRXed grain (GOS < 2°) from the unDRXed (GOS > 2°) by EBSD statistics; the obtained results and the corresponding (0001) pole figures are shown in Figure 8. It can be clearly seen that as DRX percentage increases, the maximum basal pole intensity is greatly decreases from an initial 48.07 to 12.97 in the wall zone, which indicates that the significant texture intensity weakening has achieved. Meantime, the texture type is also modified from the initial basal texture to a tilted one with the basal poles inclined from LD to TD, and the most concentrated inclination angle is ~45°.

It is well known that texture formation in Mg and its alloys is usually closely related to the applied stress condition, deformation temperature, and possible twining behavior. This resulting texture in the wall of the part is a distinct shear texture, which caused by the applied shear stress in the corner of mold. Actually, its formation is closely associated with the activation of the basal slip system that causes rotation of the grain’s basal plane toward to the geometric slip plane (shear plane) of mold’s corner. Similar texture components have also been observed previously in Mg-based alloys processed by ECAP [9,31,32]. In the current deformation process, the entire basal texture evolution can be clearly described using Figure 9. Due to the unidirectional stress state, the basal texture is formed during the upsetting stage. The basal texture is significantly dispersed in the first shear zone, but there is no obvious tilt due to the lower level of shear stress and the positive pressing force at the bottom zone. When passing through the second shear zone, the c-axis of most grains is greatly rotated toward the shear plane under higher shear stress, and thus leads to the formation of inclined basal texture in the wall zone.

On the other hand, as it can be seen in Figure 8a,c,e, the evolution of texture intensity and the appearance of texture weakening can be clearly represented by discrete (0001) pole figures. The strong basal texture (red area) appearing at zone A is consistent with the presence of a high fraction of deformed grains. With further shear deformation, DRXed grains appear and the deformed grains are greatly consumed. Meanwhile, the strong basal texture evolves into a weakly tilted DRX texture (blue area) that is composed of DRXed grains. It is obvious that these DRXed grains tend to exhibit a relatively dispersed orientation, which is the main reason for the weakening of the deformation texture. As has been observed in the evolution of microstructure, the appearance of the relatively dispersed oriented DRXed grains is actually caused by the significant CDRX and PSN effect during deformation process. Particularly for the finer DRX grains, which appear at deformed grain boundaries or are caused by PSN effect, their orientations are usually more random [33]. Besides, the fine β-particles appearing in current deformation are also an important factor contributing to the texture weakening. It has been widely reported that the appearance of fine phases in Mg-based alloys tend to possess a strong ability to prevent grain rotation, thus recreating deformation texture [11]. On the one hand, the fine phases distributed at the grain boundaries can effectively retard dislocation recombination and grain boundary migration, and thus coalescence is retarded. On the other hand, the fine phases located at grain boundaries can effectively hinder grain rotation because of their characters and high hardness, which avoids the grain sliding in the same direction, and conducive to texture modification [34]. In current study, it can be seen that the DRXed grains also tend to rotate towards the shear plane when the material passes through the second shear zone. However, due to the hindrance of fine particles, their orientation becomes less concentrated in a certain direction, which consolidates the texture weakening effect.

### 3.5. Mechanical Properties

#### 3.5.1. Microhardness

The microhardness of the longitudinal section along the deformation path of the extruded part is presented in Figure 10a. It can be seen that after the complete deformation, the hardness is significantly increased from initial ~55 HV to ~87 HV. That is, the hardness displays a more than 58% improvement under the current effective strain of ~3.3. It is worth noting that the hardness reaches a maximum (~89 HV) after passing through the first shear zone, while no increase occurs in the subsequent deformation, but decreases slightly to ~87 HV. It is clear that the results are inconsistent with the current process of grain refinement. The grain structure of the zone B and the zone C differs in that the zone B contains a large number of coarse deformed grains, but zone C is a relatively complete DRX structure. Actually, the interior of deformed grains usually has high density dislocations, thus resulting in higher hardening ability [35]. According to the work of Sabbaghianrad et al. [36], the hardness can be calculated by the modified Hollomon equation:HV = Kε_eq_^η^,(1)
where K is the materials constant, ε_eq_ is the true strain, and η is the hardenability index as the reference of the strain hardening level. Therefore, the high fraction of deformed grains (the increased strain hardening level) together with the fine DRXed grains will increase the hardness of zone B more effectively. With the further shear deformation applied in the second shear zone, the deformed grains are continuous consumed and the coarse DRXed grains appear subsequently. Grain refinement promotes hardness increase, while the decrease of dislocation density reduces hardening ability. The grain size effect is balanced by the effect of the decrease in dislocation density, which may result in no increase in hardness in the final stage.

#### 3.5.2. Tensile Properties

The engineering stress–strain curves of tensile specimens obtained from TD and LD of the wall zone are shown in Figure 10b. As it is seen, each curve includes only the strain hardening stage and the subsequent steady state stage, which is a typical stress–strain curve. The ultimate tensile strength (UTS), yield strength (YS), and elongation (EL) to failure of LD specimens are ~268.1 MPa, ~180.4 Mpa, and ~9.2%, respectively, and these values of TD specimens are further improved to ~281.3 MPa, ~189.0 Mpa, and ~10.5%, respectively. That is, the YS and EL in both directions of the extruded part increased by more than 2.5 times, and the UTS increased by more than 2 times, compared to the initial as-cast state. Comparing the mechanical properties of the two directions, it can be seen that the UTS of TD is ~13.2 MPa higher than that of LD, the YS is ~8.6 MPa higher and the elongation is ~1.3% higher. However, these slight differences in properties are not enough to cause the mechanical anisotropy of the extruded part. It is well known that the texture type and intensity of Mg alloy materials usually greatly affect the mechanical anisotropy (YS) of materials due to the basal slip and tension twining dominant room temperature plastic deformation. Due to the formation of strong texture components during plastic deformation, Mg alloy materials tend to exhibit obvious anisotropy at room temperature, such as tension–compression YS asymmetry or tension YS anisotropy [37]. Compared with traditional BE method [38], it can be seen that the texture intensity of this extruded part has been greatly weakened by introducing high effective strain and shear strain in current novel BE process. Obviously, the weakened DRX texture does not significantly cause YS anisotropy of the extruded part. Regarding the fact that the YS of TD is slightly higher than LD, this may be due to the appearance of weak shear texture, which makes the SF_basal_ along the LD larger than that along the TD, leading to lower YS along the LD (Figure 11).

The results demonstrate that the new SPD method, which successfully combines conventional BE and two-step ECAP into a single-pass forming process, can improve the properties of the AZ80 Mg alloy. In particular, the YS and El were increased by more than 2.5 times. The significant improvement in mechanical properties is inevitably related to the modified microstructure, i.e., grain, phase, and texture. Firstly, grain refinement is an important factor in simultaneously improving the material strength and plasticity. It is well known that the effect of grain size on material strength follows the Hall–Petch relation:σ_y_ = σ_0_ + Kd^−1/2^,(2)
where σ_y_ is the YS, σ_0_ is the material constant, K is Hall–Petch slope, and d is the average grain size [39,40]. However, for the K value of HCP structure materials (K_Mg_ = 280–320 MPa μm^−1/2^), it is much higher relative to materials with body-centered cubic (BCC) or face-centered cubic (FCC) structures. Usually it is more than 4 times that of Al alloy (K_Al_ = 60 MPa μm^−1/2^) [41,42]. Therefore, grain refinement is the key factor to improve the strength of Mg alloys. Compared with the coarse grain structure of as-cast materials, the average grain size of the extruded part has been greatly refined to ~9.6 μm. The significantly enhanced YS can be attributed primarily to drastic grain refinement. On the other hand, the finer the grains, the greater the number of grains per unit area, so the same plastic strain amount can be dispersed into more grains during deformation, resulting in more uniform deformation and a lower degree of stress concentration. Also, the finer the grains, the larger the grain boundary area included, and the more the grain boundaries are bent, which is more detrimental to crack propagation [4]. As a result, grain refinement also enhances the material plasticity.

Besides, the improved strength is also considered to be closely related to the fine β-particles strengthening. In general, the phase strengthening results from the ability of phases to impede dislocations motion by forcing dislocations to either cut through or get round the phases. In either case, the higher fraction of the phase leads to the higher material strength. For Mg-based alloys, phase strengthening usually follows the dislocation get round mechanism, which is also known as the Orowan mechanism [43]. Unlike the eutectic β-phase of the network morphology in the as-cast microstructure, large stress concentrations are usually caused. It can be considered that, in the postprocessed part, relatively dispersed fine β-particles are distributed at the grain boundaries and can effectively hinder the migration of dislocations and the deformation of the matrix, thereby increasing the material strength [26]. Finally, it should be considered that the appearance of texture weakening is of importance, which produced a large number of grains with relatively random orientation and soft orientation are favorable for basal slip at room temperature and, therefore, also has a certain improvement in material plasticity [5,6].

## 4. Conclusions

In this study, the annular channel angular extrusion process, as a novel SPD method, was successfully applied to fabricate an AZ80 Mg alloy cup-shaped part at 300 °C, and the microstructure, texture, and mechanical properties at room temperature were investigated. The obtained results are summarized as follows.(1)By introducing a high magnitude of effective strain, the annular channel angular extrusion process exhibited significant microstructure refinement capabilities. The coarse eutectic β-phases formed during the solidification were greatly refined into numerous fine β-particles. Due to the notable DRX nucleation and particles’ pinning effect, the average grain size of the extruded part was eventually refined to ~9.6 μm.(2)The strong basal texture formed during the initial stage was greatly weaken and transformed into a tilted DRX texture. The shear strain applied in the corner zone and subsequent microstructure refinement were the main reasons for the texture modification.(3)The material processed by the novel BE process showed simultaneous improvement of strength and ductility. The YS and EL of the extruded AZ80 alloy cup were improved more than 2.5 times, and the UTS was improved more than 2 times, compared to the as-cast state values. Besides, the obtained hardness also exhibited significant improvement.(4)The notable microstructure refinement and texture weakening were considered to be responsible for the improved mechanical properties.(5)According to this study, the SPD technique combining conventional BE and two-step ECAP into a single process can feasibly improve the mechanical properties of the AZ80 Mg alloy.

## Figures and Tables

**Figure 1 materials-12-01001-f001:**
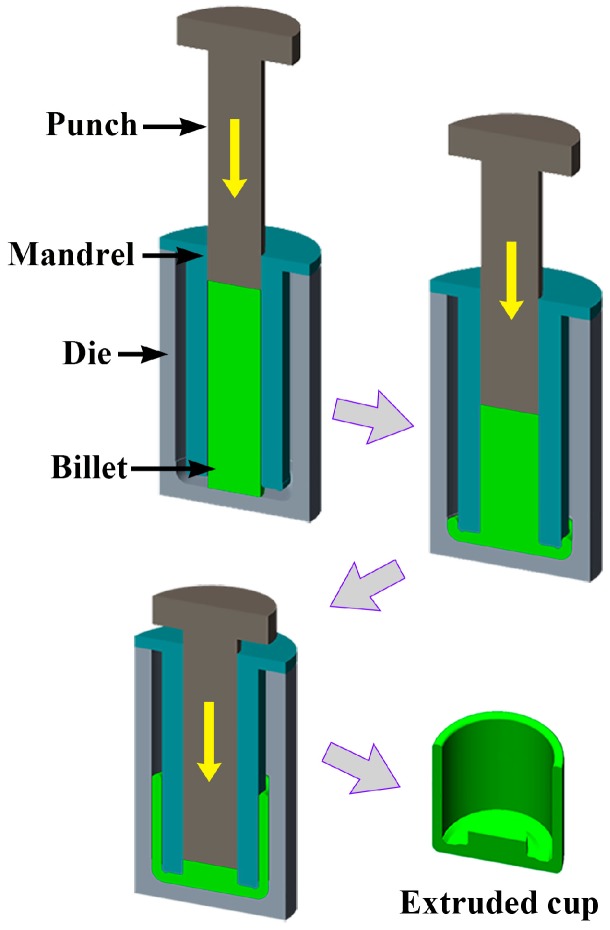
Schematic view of the annular channel angular extrusion applied in current research.

**Figure 2 materials-12-01001-f002:**
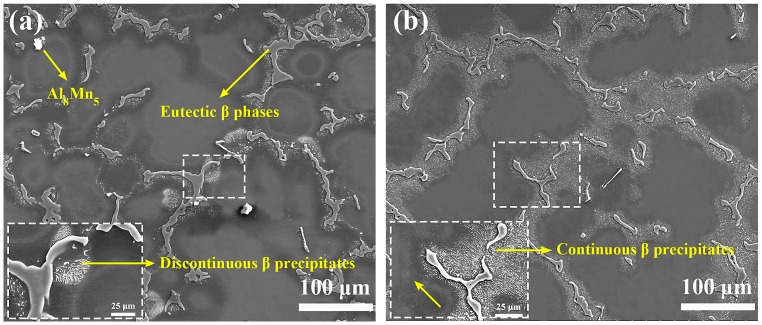
Scanning electron microscopy (SEM) microstructure of (**a**) as-cast AZ80 alloy and (**b**) billet reheated at 300 °C for 3 h.

**Figure 3 materials-12-01001-f003:**
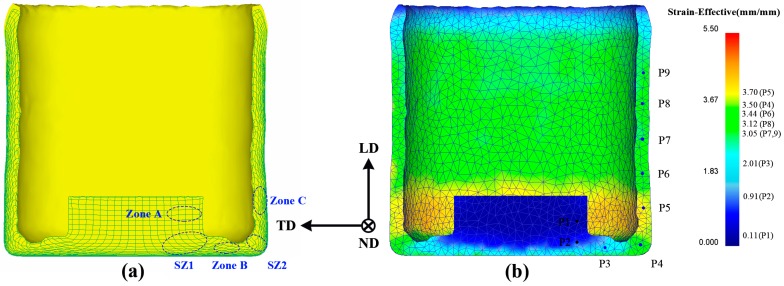
(**a**) The flow grid distribution and (**b**) effective strain distribution of the extruded part (TD, transverse direction; ND, normal direction; LD, longitudinal direction).

**Figure 4 materials-12-01001-f004:**
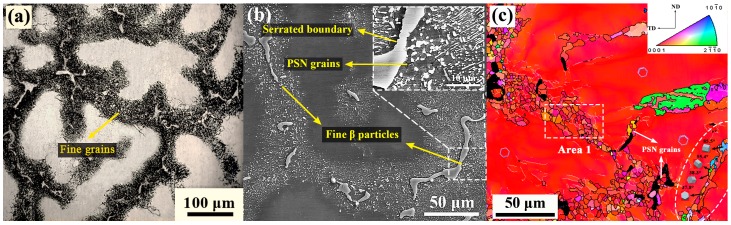
Microstructure of zone A: (**a**) optical microscopy OM, (**b**) SEM, and (**c**) electron backscattering diffraction apparatus (EBSD), where the color key triangle shows the crystallographic orientation of the hexagonal close-packed (HCP) crystal.

**Figure 5 materials-12-01001-f005:**
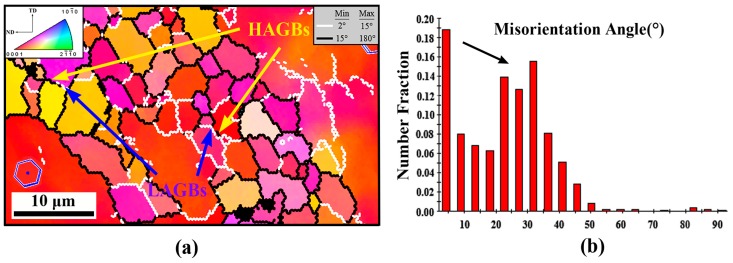
(**a**) Higher magnification of Area 1 indicated by white dashes in Figure 4c, and (**b**) corresponding misorientation angle distribution. The white and coarse black lines depict LAGBs (2°–15°) and HAGBs (15°–180°), respectively.

**Figure 6 materials-12-01001-f006:**
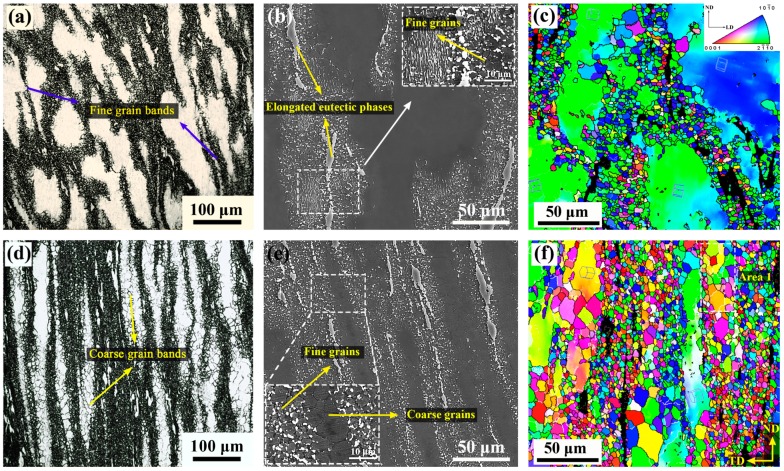
Microstructure of (**a**–**c**) zone B and (**d**–**f**) zone C: (**a**,**d**) OM, (**b**,**e**) SEM, and (**c**,**f**) EBSD, where the color key triangle shows the crystallographic orientation of HCP crystal.

**Figure 7 materials-12-01001-f007:**
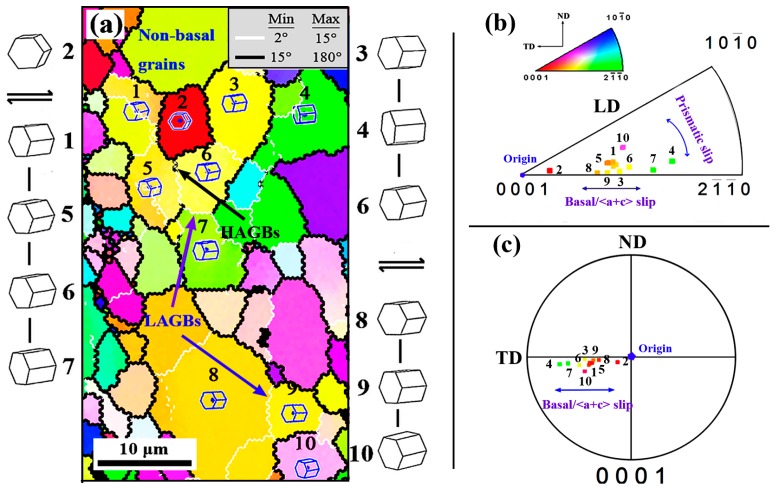
(**a**) Higher magnification of Area 1 indicated by white dashes in Figure 6f, (**b**) corresponding discrete inverse pole figure, and (**c**) discrete (0001) pole figure.

**Figure 8 materials-12-01001-f008:**
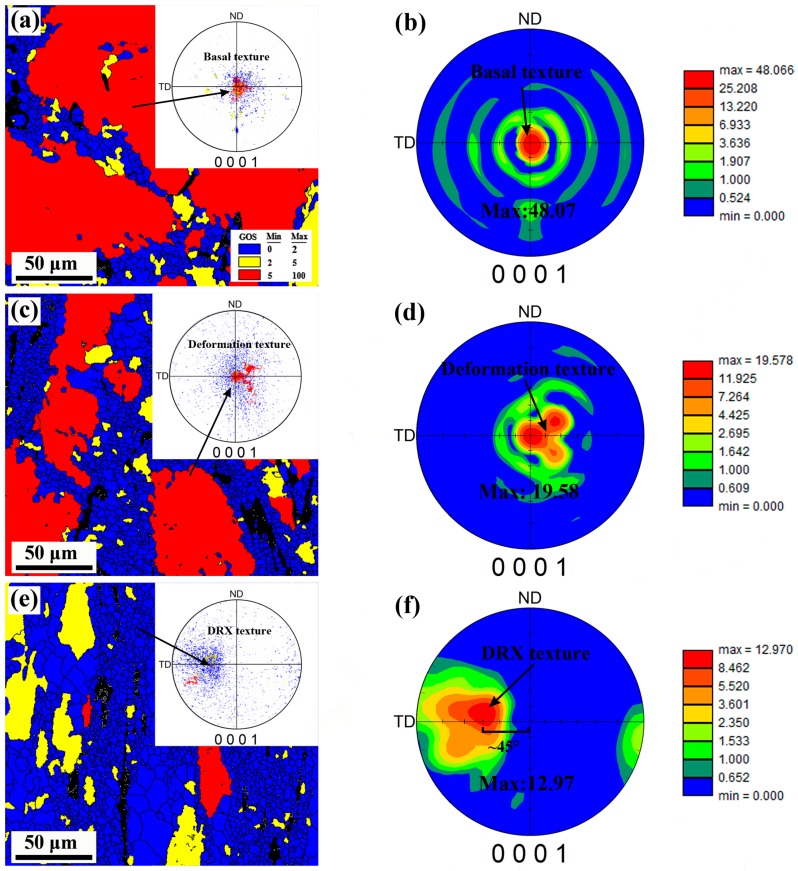
(**a**,**c**,**e**) EBSD maps distinguishing the full DRX zone (blue area), unDRX zone (yellow and red areas), and (**b**,**d**,**f**) (0001) pole figure from EBSD data.

**Figure 9 materials-12-01001-f009:**
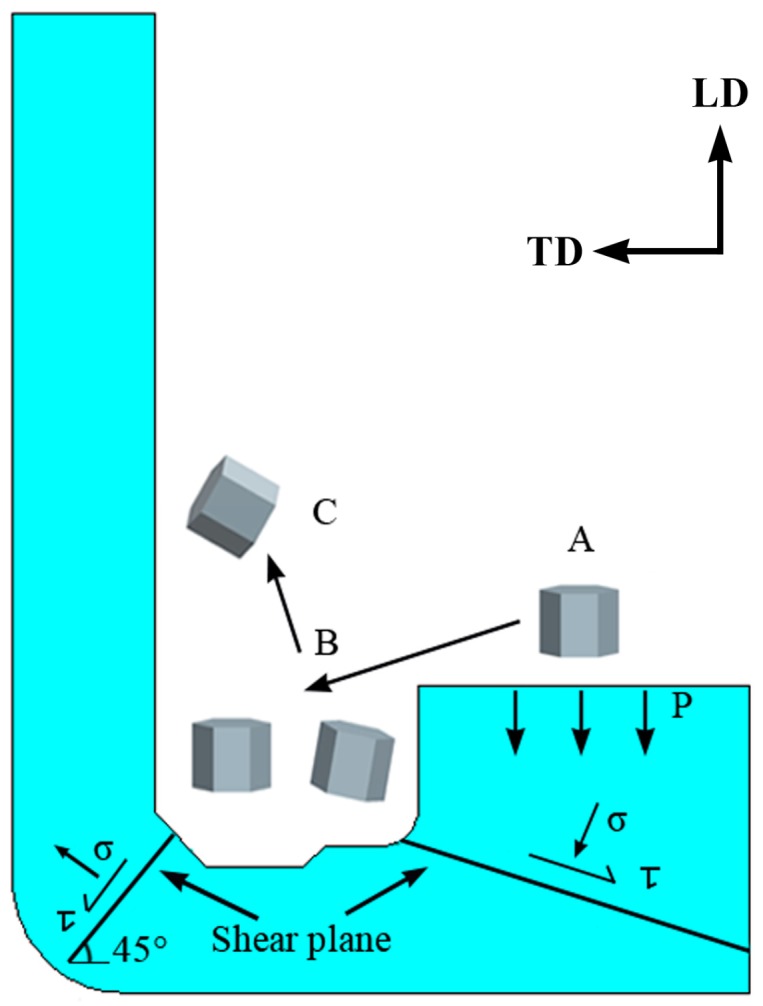
Schematic diagram of the crystal preferred orientation in the present deformation process.

**Figure 10 materials-12-01001-f010:**
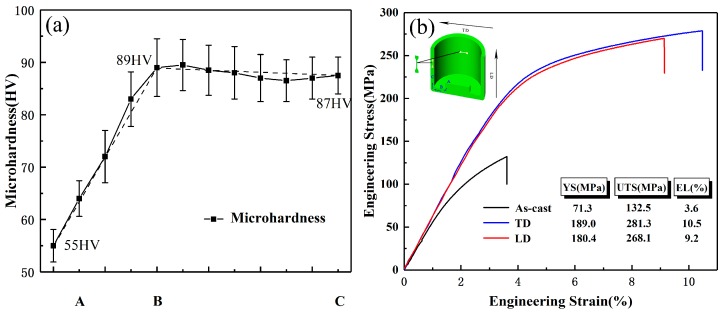
(**a**) Microhardness distribution along the deformation routine and (**b**) tensile stress–strain curves obtained from LD and TD specimens.

**Figure 11 materials-12-01001-f011:**
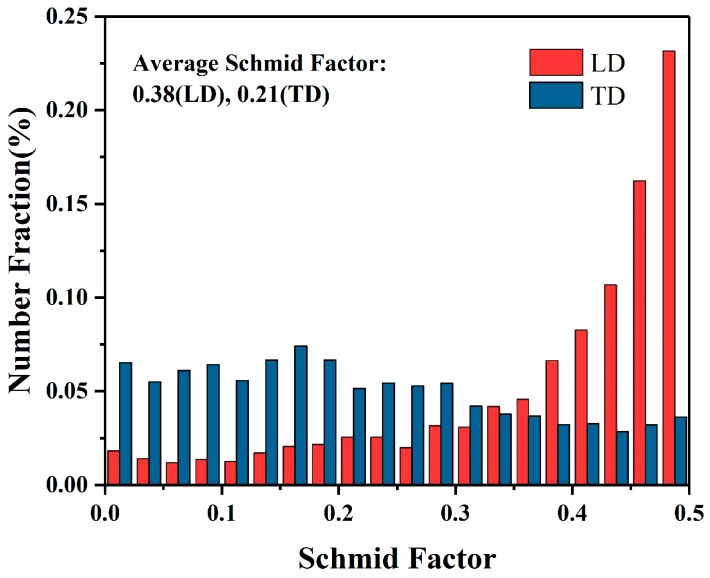
Schmid factor distribution of (0001) <11-20> basal slip in the tension directions (LD or TD) obtained from EBSD data.

**Table 1 materials-12-01001-t001:** The parameters of the simulation.

Simulation Parameters	Values
Elastic modulus (GPa)	45.0
Poisson’s ratio	0.35
Height of billet (mm)	360
Diameter of billet (mm)	90
Mesh number of billet (1/2)	32,000
Minimum mesh edge length (mm)	2.15
Friction coefficient	0.3
Extrusion temperature (°C)	300

**Table 2 materials-12-01001-t002:** The dynamic recrystallized (DRXed) grain size, average grain size, dynamic recrystallization (DRX) fraction, and relative fraction of eutectic phase to fine particle of zones A, B, and C, respectively.

Zone	d_DRXed_ (μm)	d_average_ (μm)	f_DRX_ (%)	f_eutectic phase_ (%)	f_fine particle_ (%)
A	6.1 (±1.1)	_	23 (±5)	89 (±5)	11 (±3)
B	5.3 (±1.3)	_	53 (±4)	65 (±4)	35 (±4)
C	8.3 (±0.9)	9.6 (±1.4)	85 (±2)	39 (±4)	61 (±3)

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
