# Peer review of "An Investigation on Microstructure, Texture and Mechanical Properties of AZ80 Mg Alloy Processed by Annular Channel Angular Extrusion"

_materials, 2019, doi:10.3390/ma12061001_

Reviewer 1 Report

Review for materials- 472320

An Investigation on Microstructure, Texture and Mechanical Properties of AZ80 Mg Alloy Processed by Annular Channel Angular Extrusion

The authors address an interesting research topic for the journal Materials. It is a rigorous and well-organized paper. Anyway, some recommendations should be considered:

For the sake of clarity, please indicate in the main text if a mesh convergence study was carried out for FE analysis.

The figure 5 should be enlarged for better understanding.

Please, Vikers hardness should be defined as HV instead Hv.

Figure 10: MPa instead Mpa.

Although TD and LD is indicated in a figure, please define these acronyms in the main text.

Although the number and the selection of references is adequate, it would be advisable to include some papers from the journals of MDPI editorial (Materials, Metals, etc.) related to the topic of the manuscript.

Author Response

Dear Sir or Madam:

Thank you very much for your attention and the comments on our paper. Based on your comment and request, we have made modification on the original manuscript. The following is the result of our revision, please check:

Point 1: For the sake of clarity, please indicate in the main text if a mesh convergence study was carried out for FE analysis.

Response 1: The mesh convergence analysis has been added to the “Materials and Methods section of the article (Section 2, line 123). Since the too fine mesh will affect the visibility of the image, in order to observe the picture more clearly, the image we use is the result of the finite element simulation just emerging convergence.

Point 2: The figure 5 should be enlarged for better understanding.

Response 2: Figure 5 has been modified, please confirm (line 222).

Point 3: Please, Vikers hardness should be defined as HV instead Hv.

Response 3: All “Hv” in this article, including picture (Figure 10a) and texts, have been replaced by “HV”. Please verify.

Point 4: Figure 10: MPa instead Mpa.

Response 4: The “Mpa” in Figure 10b has been replaced by “MPa” (line 380).

Point 5: Although TD and LD is indicated in a figure, please define these acronyms in the main text.

Response 5: According to the first appearance of TD, LD, and ND (line 175), we defined them. Please confirm.

Point 6: Although the number and the selection of references is adequate, it would be advisable to include some papers from the journals of MDPI editorial (MaterialsMetals, etc.) related to the topic of the manuscript.

Response 6: According to your suggestion, we have replaced some references, which are respectively: [1], [2], [4], [6], [9] and [31]. Please confirm.

Best regards

Sincerely yours

Xi Zhao

Reviewer 2 Report

In this research work the Mg alloy AZ80 is submitted to annular channel angular extrusion for refining its microstructure. It is carried out a study of the microstructure by optical microscopy, scanning electron microscopy and EBSD. The mechanical properties are also investigated from microhardness measurements and tensile tests.

The study presented in the manuscript is quite complete, original and new, in particular the method used for refining the microstructure. The manuscript deserves to be published after minor changes.

1.  In the lines 36-37 it is said: “casting Mg alloys have been gradually can no longer meet the practical applications”. There is something missing in the sentence, maybe after “gradually”?  The sentence cannot be understood.

2.In the line 67 it is said: “the novel process could obtained more than twice..” Is it “could be obtained more than twice..”?

3.  In the line 88. The dimensions of the sample after the extrusion process are given. The authors should give the outer and inner diameters of the cup.

4.   In the line 107. The authors give the dimensions of the tensile specimens. They should also give the thickness

5. In the lines 132-133. It is said: “This certain orientation relationship has also been reported in depth by many studies..”. The authors should give at least a reference in which these results have been reported.

6.  In the lines 141-151. The authors are commenting a figure, I think that it is figure 3, but the reference is not given in the text. The authors should indicate that they are speaking about figure 3  (probably figure 3a).

7.  In the lines 190-191. It is said: “However, they are located on the grain boundaries may become barriers to grain ….” There is something missing in the sentence and it cannot be understood.

8. In the lines 197-202. It is said that the figure 4c also shows that slight twin dynamic recrystallization has also taken place within the deformed grains. The authors should indicate with arrows in the figure where the twins are observed.

9.  In the line 231. It is said: “The material tends to cannot withstand the applied shear”.  Tend to cannot, is it correct in english?

10. In the line 367, It is said “As is seen, the ultimate tensile..” is it not better “As it is seen?

11. In the section 3.5.2, Tensile properties. The authors devote most part of this section to explain the increase of the yield strength obtained in the tensile tests when the tension is applied along the LD or TD directions with respect to the as cast specimens. However, another important point is the increase in the ductility which is observed in the tensile curves. This aspect is hardly commented and only a small sentence is given and it is referenced to ref. [5]. The authors should comment a little more this part of the tensile curves and try to give some explanations to the increase in the plastic deformation with respect to the as cast specimens.

Author Response

Dear Sir or Madam:

Thank you very much for your attention and the comments on our paper. Based on your comment and request, we have made modification on the original manuscript. The following is the result of our revision, please check:

Point 1: In the lines 36-37 it is said: “casting Mg alloys have been gradually can no longer meet the practical applications”. There is something missing in the sentence, maybe after “gradually”?  The sentence cannot be understood.

Response 1: I'm very sorry for my mistake in English writing. The sentence has been revised to: “Cast Mg alloy products have gradually been unable to meet the practical application needs” (line 36).

Point 2: In the line 67 it is said: “the novel process could obtained more than twice..” Is it “could be obtained more than twice..”?

Response 2: The sentence (line 68) has been modified, please check.

Point 3 and 4: In the line 88. The dimensions of the sample after the extrusion process are given. The authors should give the outer and inner diameters of the cup. In the line 107. The authors give the dimensions of the tensile specimens. They should also give the thickness

Response 3 and 4: I am sorry that the detailed parameters of the extruded part and the thickness of the tension specimen have not been explained. The data have been increased to lines 90 and 112, respectively. Please verify.

Point 5: In the lines 132-133. It is said: “This certain orientation relationship has also been reported in depth by many studies..”. The authors should give at least a reference in which these results have been reported.

Response 5: Reference [18-20] has been moved to this position (line 141), please confirm.

Point 6: In the lines 141-151. The authors are commenting a figure, I think that it is figure 3, but the reference is not given in the text. The authors should indicate that they are speaking about figure 3 (probably figure 3a).

Response 6: This text is indeed described in Figure 3a, we have made the corresponding changes, please verify.

Point 7: In the lines 190-191. It is said: “However, they are located on the grain boundaries may become barriers to grain ….” There is something missing in the sentence and it cannot be understood.

Response 7: The sentence has been revised to: “Their presence at the grain boundaries may hinder the migration and coalescence of grains by pinning effect, and thus can greatly delay the growth of newly DRXed grains” (line 203).

Point 8: In the lines 197-202. It is said that the figure 4c also shows that slight twin dynamic recrystallization has also taken place within the deformed grains. The authors should indicate with arrows in the figure where the twins are observed.

Response 8: According to your requirement, in Figure 4c we mark the twin grains in the twin band, please verify.

Point 9: In the line 231. It is said: “The material tends to cannot withstand the applied shear”.  Tend to cannot, is it correct in english?

Response 9: I'm very sorry for the grammatical mistake. The “cannot” has been replaced by “be unable to” (line 245).

Point 10: In the line 367, It is said “As is seen, the ultimate tensile..” is it not better “As it is seen?

Response 10: “As is seen” has been replaced by “As it is seen”.

Point 11: In the section 3.5.2, Tensile properties. The authors devote most part of this section to explain the increase of the yield strength obtained in the tensile tests when the tension is applied along the LD or TD directions with respect to the as cast specimens. However, another important point is the increase in the ductility which is observed in the tensile curves. This aspect is hardly commented and only a small sentence is given and it is referenced to ref. [5]. The authors should comment a little more this part of the tensile curves and try to give some explanations to the increase in the plastic deformation with respect to the as cast specimens.

Response 11: We have added a description of the tensile curves (line 382) and an explanation for the plasticity improvement (line 410 and 420), please confirm.

Best regards

Sincerely yours

Xi Zhao

Reviewer 3 Report

referee report 

materials-472320

Title: An Investigation on Microstructure, Texture and Mechanical Properties 

of AZ80 Mg Alloy Processed by Annular Channel Angular Extrusion

Authors: Xi Zhao *, Shuchang Li, Yong Xue, Zhimin Zhang

The extrusion process of Mg-based alloys is a very important deformation technique for industrial materials. The analysis

of the microstructure, texture and the mechanical properties of AZ80 Mg alloy is well suited for materials.

The manuscript is well planned and well written, and a lot of efforts was invested to prepare the figures. In the attached document, some problems are shown and some advices are given.

Another small remark:

There is always a space between a word and a following bracket. Please introduce them in the text, the figure captions and in the axis labels in the figures.

Author Response

Dear Sir or Madam:

Thank you very much for your attention and the comments on our paper. Based on your comment and request, we have made modification on the original manuscript. The following is the result of our revision, please check:

Response 1: In the line 68, a space has been added between the “finite element” and the “(FE)”. Please confirm.

Response 2: In the line 103, “the OIM 7.3 software” has been defined as: “Orientation Imaging Microscopy (OIM) Analysis-v7.3 software”.

Response 3: The spaces have been added in the corresponding position in Table 1, please check.

Point 4: Figures 4a and 4b are not referred to in the text, only generally as Fig. 4.

The immediate question if the area of 4b equals 4c remains unanswered

Response 4: “Figure 4” in the line 180 has been replaced by “Figure 4a, b and c”.

Point 5:  The blue arrows (Figure 5) are unmotivated and confusing. They are also not mentioned in the figure caption. Better remove them. Besides, this figure is too small to be read. Make it larger aside the image.

Response 5: The arrows in the Figure 5 refer to LAGBs and HAGBs, which we have adjusted and explained in the text (lines 188-189). In addition, the picture has also been adjusted accordingly, please check.

Point 6:  The blue arrows (Figure 7) have here the same problems as before.

Again, the arrows are not mentioned in the figure caption.

Response 6: The arrows in the Figure 5 refer to LAGBs and HAGBs, which we have adjusted and explained in the text (line 277). Please check.

Point 7: In Figure 10b, the schematic diagram could be enlarged -- there is so much space unused

Response 7: The image has been adjusted accordingly, please check.

Best regards

Sincerely yours

Xi Zhao